# Reptile Identification for Endemic and Invasive Alien Species Using Transfer Learning Approaches

**DOI:** 10.3390/s24051372

**Published:** 2024-02-20

**Authors:** Ruymán Hernández-López, Carlos M. Travieso-González

**Affiliations:** Signals and Communications Department (DSC), Institute for Technological Development and Innovation in Communications (IDeTIC), University of Las Palmas de Gran Canaria (ULPGC), 35017 Las Palmas de Gran Canaria, Spain; ruyman.hernandez@ulpgc.es

**Keywords:** transfer learning, deep learning, wildlife recognition, animal identification, Canarian endemic species, invasive alien species, biodiversity conservation, TensorFlow, Keras

## Abstract

The Canary Islands are considered a hotspot of biodiversity and have high levels of endemicity, including endemic reptile species. Nowadays, some invasive alien species of reptiles are proliferating with no control in different parts of the territory, creating a dangerous situation for the ecosystems of this archipelago. Despite the fact that the regional authorities have initiated actions to try to control the proliferation of invasive species, the problem has not been solved as it depends on sporadic sightings, and it is impossible to determine when these species appear. Since no studies for automatically identifying certain species of reptiles endemic to the Canary Islands have been found in the current state-of-the-art, from the Signals and Communications Department of the Las Palmas de Gran Canaria University (ULPGC), we consider the possibility of developing a detection system based on automatic species recognition using *deep learning (DL)* techniques. So this research conducts an initial identification study of some species of interest by implementing different neural network models based on transfer learning approaches. This study concludes with a comparison in which the best performance is achieved by integrating the *EfficientNetV2B3* base model, which has a mean *Accuracy* of 98.75%.

## 1. Introduction

An ecosystem is a complex biological system characterized by both biotic components, forming a community of living organisms, and abiotic components, comprising the non-living elements present in the natural environment. Together, these components operate as a cohesive unit. However, when a species transcends biogeographical barriers and enters a new region, it can disrupt the delicate balance of the ecosystem. This disruption manifests as alterations to ecosystem functioning and the provision of ecosystem services and impacts processes such as nutrient and contaminant cycling, hydrology, habitat structure and disturbance regimes. Invasive alien species (IAS) break down biogeographic realms, affect native species richness and abundance, increase the risk of native species extinction, affect the genetic composition of native populations, change native animal behaviour, alter phylogenetic diversity across communities and modify trophic networks [1].

The Canary Islands are considered a hotspot of biodiversity [2], and the high diversity of habitats, geological isolation from any major landmass, interspecific competition and adaptive radiation are some of the causal factors that have been suggested to explain the high levels of endemicity found in this archipelago [3]. The reptiles inhabiting the Canary Islands form a distinct group of 15 living species characterized by well-defined insular distributions. Among these, 14 species are endemic and exhibit limited capacity to disperse across marine barriers. Notably, the distribution pattern of these endemic reptiles includes the sharing of several islands by the same species [4] where some native herpetofauna species are considered endangered.

Currently, some IAS of reptiles are proliferating with no control in different parts of the territory, creating a dangerous situation for the ecosystems of this archipelago. The introduction of invasive species to islands, coupled with the loss and fragmentation of natural habitats, constitutes one of the most severe threats to the conservation of biological diversity. Furthermore, the vulnerability to invasion is significantly heightened in the Canary Islands due to the distinctive ecological conditions under which island organisms have evolved. In other words, the absence of adaptations to predators, low genetic diversity and increased susceptibility to exotic pathogens, among other factors, amplify the detrimental effects of biological invasions in the Canary Islands compared to continental ecosystems [5]. Nevertheless, efforts to mitigate this problem encounter numerous obstacles given that it involves a complex interplay of technical, political, economic and social aspects. The multifaceted nature of the issue transcends the jurisdiction of a single administration or even a single country. The challenge in implementing barriers to free trade among European Union member countries, coupled with the impracticality of comprehensive surveillance to prevent the introduction and release of species, significantly constrains the possibilities for effective action in this regard [6].

The regional authorities have taken proactive measures to control the proliferation of invasive species, exemplified by the establishment of the Canary Islands Early Warning Network for the Detection and Intervention of Invasive Alien Species, known as RedEXOS (La Red de Alerta Temprana de Canarias para la Detección e Intervención de Especies Exóticas Invasoras) [5]. The management strategies employed rely on an information system designed for monitoring invasive alien species in the Canary Islands. This system functions as an administrative communication mechanism and hinges on the voluntary participation of individuals who report the presence of specimens when sightings occur. Nevertheless, the issue persists, as the population sizes of certain invasive species remain uncertain, and there is an ongoing threat to native species, as the invasive species continue to jeopardize the ecological balance.

Given the impossibility of precisely determining the times at which these species appear and recognizing that the warnings issued through the Canary Islands Early Warning Network rely on sporadic sightings by volunteers, from the Signals and Communications Department of the Las Palmas de Gran Canaria University, we consider the possibility of developing a detection system based on automatic species recognition using *deep learning (DL)* techniques with the aim of enhancing the efficiency of monitoring and controlling the relevant species in the Canary Islands.

To address the challenges posed by the absence of sightings or the reported presence of relevant species, we propose the implementation of an automatic monitoring system. This system would utilize volumetric motion sensors and camera traps to detect and record the presence of species more proactively and continuously. Volumetric motion sensors can be employed to activate camera traps upon detecting the presence of species in predefined spaces. The integration of these motion sensors with camera traps allows for the automated recording of images when triggered by the detected motion. This particular camera type provides the capability for continuous real-time monitoring. By incorporating automatic identification algorithms, the system can promptly activate an alert signal that notifies system administrators of the presence of any of the relevant species. The identification of species through strategically placed cameras in the Canary Islands will empower regional authorities to implement targeted measures. This includes actions like capturing the identified specimens and providing improved ecosystem monitoring.

To initiate the research, this paper suggests conducting an initial identification study focused on certain species of faunistic interest using various classification models. Two of these species are documented as having particular significance in the state catalogue, while the other two are included in the Spanish catalogue of invasive alien species.

### 1.1. Related Work

To know that the most relevant techniques are being employed, a study on the state-of-the-art has been conducted. The current state-of-the-art for automatic recognition based on computer vision encompasses numerous studies focused on species identification, and among the commonly used techniques for both species and individual identification, the following can be found in this review [7]:Support Vector Machine (SVM);Scale-Invariant Feature Transform (SIFT);Ensemble of Exemplanar Support Vector Machine (EESVM);Random Forest Algorithm;Optical Flow;*k*-nearest neighbour classifier (*k*NN).

On the other hand, in the domain of deep learning, the following can be highlighted:Convolutional Neural Network (CNN);Recurrent Convolutional Network (RCN);VGG-Face Convolutional Neural Network;Deep segmentation convolutional neural network;You Only Look Once (YOLO).

Deep learning relies on multilayered, connected processing units called *Artificial Neural Networks (ANNs)*, and this subset of *machine learning (ML)* techniques is at the core of emerging technologies such as self-driving cars and is responsible for significant improvements to widely used information technology tools such as image and speech recognition and automated language translation [8]. However, in comparison to traditional machine learning techniques, deep learning has surpassed the state-of-the-art in the realm of detecting wildlife species [9].

As an example, in this study [10], WilDect-YOLO, a deep learning (DL)-based automated high-performance detection model for real-time endangered wildlife detection was developed and obtained a mean average *Precision* value of 96.89%.

In addition, automatic detection can be applied to accurately count the number of animals in a herd, as demonstrated in this research [11] in which various types of CNNs were implemented to achieve precise detection and counting of African mammals through analysis of aerial imagery. Even in scenarios demanding the monitoring of expansive populations of terrestrial mammals, a combination of satellite remote sensing and deep learning techniques can be employed [12].

Furthermore, the Transfer Learning method can be effectively combined with some of the aforementioned techniques. Indeed, Transfer Learning has found application in various works across diverse domains. In the realm of wildlife identification, this method has been utilized for fish identification in tropical waters [13], distinguishing between different dog breeds [14] and accurately identifying various bird species [15].

Regarding the species under study, the state-of-the-art showcases a diverse range of research efforts focused on the identification of various species. Numerous works in the field have successfully identified different snake species, as exemplified by this work [16], and there is even some other research in which different species of herpetofauna can be recognised, such as [17].

### 1.2. Contributions

The research presented in this paper has been conducted with the goal of automatically classifying images of various species, including both invasive alien species and endemic species, found in the Canary Islands through computer vision techniques.

The conceptual schematic diagram of the work carried out is given in Figure 1.

As can be seen in the outline, the research methodology involves several key steps. Initially, a database is curated, comprising images of the species under investigation. During this process, the samples are meticulously labelled, with each species assigned to a distinct class. Once the database is compiled, various classification models are implemented. Subsequently, the samples are input into these models to undergo classification. The outcomes of the classification process are then evaluated using different metrics. Finally, a comprehensive comparison of the various models employed in the study is conducted based on the evaluation metrics to assess and rank their performance. This structured approach ensures a thorough and systematic analysis of the effectiveness of the implemented classification models.

The novelty of this study, in comparison to the existing state-of-the-art, resides in the specific focus on the types of species being classified. While numerous studies exist for various species globally, such as fish, mammals, birds and herpetofauna, there is a distinctive gap in the literature when it comes to applying deep learning techniques for discriminating between species endemic to the Canary Islands and IAS introduced into this archipelago.

This research stands out as a pioneering effort for addressing the unique ecological context of the Canary Islands, where both endemic and invasive species coexist. By applying deep learning techniques to this specific scenario, the study aims to contribute novel insights into the automated classification of species within this distinct geographical and ecological setting. This targeted focus enhances the significance and originality of the research in the broader context of species classification using deep learning methods.

Hence, the purpose of this research is to conduct an initial approach, applying deep learning techniques commonly used in species identification, to identify relevant species in the Canary Islands so as to be able to monitor the ecosystems of this archipelago more efficiently. This research is dedicated to exploring the effectiveness of applying various models for the classification of species in images, with a specific emphasis on discerning which models yield the most favourable results.

This paper is structured as follows: First, a section on materials and methods will provide an overview of the materials used and the methodology employed in the research. Following this, the experimental methodology will elaborate on the experimental procedures and detail how the experiments were conducted. Then, there is a section dedicated to the results obtained from the experiments. And finally, the discussion section will delve into an analysis and interpretation of the results and offer insights and implications arising from the research.

## 2. Materials and Methods

This section introduces the samples constituting the database and evaluates their relevance for this research. Additionally, it provides an overview of the deep learning techniques and models that have been implemented in the study.

### 2.1. Datasets and Data Selection

In the domain of pattern recognition, the presence of a well-suited learning dataset is pivotal. The training dataset, derived from the original dataset, plays a central role in training, evaluating and ultimately constructing the classifier.

Nowadays, access to diverse public databases facilitates the acquisition of images depicting various species. However, for the purposes of this study, a specific database has been meticulously curated. This database is constructed from images of four species sourced from different websites on the internet. The deliberate compilation of this custom database allows for a targeted and controlled dataset that is tailored to the specific objectives of the research.

The selected reptile species included in the database are of ecological significance as they either inhabit or have been observed on Gran Canaria island. Specifically, two of these species are documented as having particular ecological importance and are registered as such in the state catalogue [18]: the Gran Canaria giant lizard, *Gallotia stehlini (Schenkel, 1901)* and the Gran Canaria skink, *Chalcides sexlineatus (Steindachner, 1891).* The remaining two species in the dataset are the Yemen chameleon, *Chamaeleo calyptratus (Duméril & Duméril, 1851)* and the ball python, *Python regius (Shaw, 1802).* Both of these species are categorized as invasive alien species of concern for the outermost region of the Canary Islands as outlined in the Spanish catalogue of invasive alien species [19].

From the original dataset, the training dataset is derived. This subset of data is employed to train, evaluate and consequently construct the classifier. In these initial experiments, the data comprising the original dataset were obtained by downloading images from various websites. The searches conducted to gather these data did not prioritize specific entities or sources. The primary focus of this study is on evaluating the discrimination capacity of *Keras* models for the classification of reptiles, specifically these species, which exhibit some degree of dissimilarity. Notably, certain data used in the study have been sourced from specialized websites such as:www.reptile-database.reptarium.cz (accessed on 1 December 2023);www.biodiversidadcanarias.es (accessed on 1 December 2023);www.inaturalist.org (accessed on 1 December 2023).

Additionally, in some other cases, images have been sourced from non-specialized websites, including platforms such as *Wikipedia* or *Flickr*.

Despite the variation in data sources, whether from specialized sites or non-specialized platforms, the samples provided exhibit a wide array of snapshots. These images encompass different perspectives, angles, and foregrounds and present the specimens in high-quality images where they can be easily distinguished by the human eye. This is the *Good Insight Dataset*, and some sample examples are shown in Figure A1 of Section A.1 (Appendix A).

Endemic species (examples shown in Figure A1a–d):-The Gran Canaria giant lizard is shown in the photos of both Figure A1a and Figure A1b, where both specimen No. 1 and specimen No. 2 have been photographed in their respective right profiles.-The Gran Canaria skink is presented in the photos of both Figure A1c, where the left profile of specimen No. 3 is shown, and Figure A1d, where a top-down perspective of specimen No. 4 is shown.Invasive alien species (examples shown in Figure A1e–h):-The Yemen chameleon is displayed in the images of both Figure A1e, where the right profile of specimen No. 5 is shown, and Figure A1f, where the left profile of specimen No. 6 is shown.-The ball python is exhibited in the photos of both Figure A1g, where a top-down perspective of specimen No. 7 is shown, and Figure A1h, where a top-down perspective of specimen No. 8 is shown.

In contrast, certain pictures depict specimens in ways that pose challenges for human recognition, or it has been anticipated that these instances might present difficulties for models to accurately classify them. To locate images under such challenging conditions, snapshots were searched using the *Google Images* website. The underlying concept is to evaluate the performance of these models not only under optimal conditions where species are easily distinguishable but also under challenging scenarios where visibility conditions are adverse. This approach aims to simulate real-world situations where recognition difficulties may arise.

In these instances, some of the samples include shots captured under unfavourable light conditions, specimens with their bodies partially obscured by objects, specimen photos captured too close to the camera or cases where more than one specimen has been photographed. This is the *Wild Dataset*, and some sample examples are shown in Figure A2 of Section A.2 (Appendix A).

Endemic species (examples shown in Figure A2a–d):-The Gran Canaria giant lizard is shown in the photos of both Figure A2a, where specimen No. 9 appears with the body out of focus, and Figure A2b, where specimen No. 10 presents a perspective from behind.-The Gran Canaria skink is presented in the photos of both Figure A2c, where only the head and the front part of the body of specimen No. 3 can be seen, and Figure A2d, where some branches partly interrupt the view of specimen No. 11.Invasive alien species (examples shown in Figure A2e–h):-The Yemen chameleon is displayed in the images of both Figure A2e, where part of the face of specimen No. 12 is focused too closely, and Figure A2f, where some branches partly interrupt the view of specimen No. 13.-The ball python is exhibited in the photos of both Figure A2g, where there are two specimens—specimen No. 8 is shown together with a new specimen: specimen No. 14—and Figure A2h, where some branches and leaves partly interrupt the view of specimen No. 15.

For the experiments in this research, a balanced dataset with 40 samples per class of the species under study was employed. The original images from which the database was comprised exhibit variations in resolution, with the number of pixels per sample ranging from 38,160 to 45,441,024. The sizes of these particular samples are specified as 240×159 and 8256×5504, respectively. To standardize the input for the classifier, all samples were resized to a uniform size of 200×200. Hence, this work utilized a total of 160 RGB images encoded as JPEGs and standardized to a resolution of 200×200 pixels. The selected samples, which constitute the dataset for this study, are summarized in Table 1.

Concerning the authenticity of the sample labels, it must be said that while verification may have been conducted on the specialized pages from which the data were downloaded, no herpetofauna experts were directly involved in this research to provide detailed identification of the species depicted in each image within the database. Nonetheless, the individuals responsible for downloading the images are natives of Gran Canaria (Canary Islands) and have been able to perfectly identify the endemic species that are the subject of this study: both the Gran Canaria giant lizard and the Gran Canaria skink. Their local knowledge and familiarity with the unique herpetofauna of the region contribute to a reliable identification process for these specific species.

In the context of this study, regarding invasive alien species, it is deemed that the downloaded samples corresponding to these species exhibit distinctive body patterns compared to others in this database. Specifically, one of them is the sole chameleon, and the other is the only snake, making all of them considered as *“ground truth”* [20] within the scope of this research. It is necessary to recall the definition of the concept, as *ground truth* is a conceptual term related to the knowledge of the truth concerning a specific question. It is the ideal expected result [21]. Ground truth or reference data are the basis for performance analysis in computer vision and image processing. This term originally stems from geography, where information drawn from satellite images is confirmed by people visiting the location to be studied on the ground [22].

### 2.2. Recognition of Species Using Deep Learning Approaches

The machine learning platform behind the classification algorithms implemented in this research is *TensorFlow v2.15.* TensorFlow is an end-to-end open-source platform for machine learning that has a comprehensive, flexible ecosystem of tools, libraries and community resources that lets researchers push the state-of-the-art in ML and developers easily build and deploy ML-powered applications [23]. The implementation of the classification models was based on *Keras* [24], which is a deep learning API (*Application Programming Interface*) written in *Python* and running on top of the machine learning platform TensorFlow [25].

While TensorFlow is an infrastructure layer for differentiable programming and deals with tensors, variables and gradients, Keras is a user interface for deep learning and deals with layers, models, optimizers, loss functions and metrics, among other factors. So Keras serves as the high-level API for TensorFlow. Keras applications are transfer learning models that are made available alongside pre-trained weights. These models can be used for prediction, feature extraction and fine tuning.

Deep learning, and specifically, Convolutional Neural Networks, have drastically improved how intelligent algorithms learn. A CNN is a class of Artificial Neural Network (ANN) that is most commonly used for image analysis and learns directly from data. In addition, with convolutional layers, pooling layers and fully connected layers, CNNs allow computational models to represent data with multiple levels of abstraction.

On the other hand, CNNs are commonly developed at a fixed resource budget and then scaled up for better *Accuracy* if more resources are available.

#### Transfer Learning and Recognition Models

The pre-trained models used in this survey apply the deep learning technique on which the classifiers implemented in this study are based, which is called *Transfer Learning.*

Many machine learning methods work well only under a common assumption: the training and test data are drawn from the same feature space and the same distribution. When the distribution changes, most statistical models need to be rebuilt from scratch using newly collected training data. In many real-world applications, it is expensive or impossible to recollect the needed training data and rebuild the models. It would be nice to reduce the need and effort to recollect the training data. In such cases, knowledge transfer or transfer learning between task domains would be desirable [26].

Transfer Learning is a machine learning method whereby a learning model developed for a first learning task is reused as the starting point for a learning model in a second learning task [27]. This is possible because of the re-use of pre-trained weights. Pre-trained weights refer to using pre-trained neural networks, which have been previously trained with some kind of data. Therefore, it can be said that learning is transferred and is available for new experiments with other types of data. Furthermore, transfer learning enables experiments to be developed with databases with few samples, such as the one available for this research. This is because some of these pre-trained models have been trained with datasets from the web containing about a million images and 1000 different classes [28].

The following is a formal explanation of the Transfer Learning technique [29]:

A domain *D* is defined by two parts: a feature space X and a marginal probability distribution P(X), where $X=\{x_{1},...,x_{n}\}\in X$, $x_{i}$ is the i−th feature vector (instance), *n* is the number of feature vectors in *X*, X is the space of all possible feature vectors, and *X* is a particular learning sample. For a given domain *D*, a task *T* is defined by two parts: a label space *Y* and a predictive function f(·), which is learned from the feature vector and label pairs $\left\{x_{i},y_{i}\right\}$, where $x_{i}\in X$ and $y_{i}\in Y$.

Taking into account that a domain is expressed as D={X,P(X)} and a task is expressed as T={Y,f(·))}, a $D_{S}$ is defined as the source domain data, where $D_{S}=\left\{\left(x_{S_{1}},y_{S_{1}}\right),...,\left(x_{S_{n}},y_{S_{n}}\right)\right\}$, where $x_{S_{i}}\in X_{S}$ is the i−th data instance of $D_{S}$, and $y_{S_{i}}\in Y_{S}$ is the corresponding class label for $x_{S_{i}}$. In the same way, $D_{T}$ is defined as the target domain data, where $D_{T}=\left\{\left(x_{T_{1}},y_{T_{1}}\right),...,\left(x_{T_{n}},y_{T_{n}}\right)\right\}$, where $x_{T_{i}}\in X_{T}$ is the i−th data instance of $D_{T}$, and $y_{S_{i}}\in Y_{S}$ is the corresponding class label for $x_{T_{i}}$. Further, the source task is notated as $T_{S}$, the target task as $T_{T}$ and the source predictive function as $f_{T}\left(\cdot \right)$.

Then, given a source domain $D_{S}$ with a corresponding source task $T_{S}$ and a target domain $D_{T}$ with a corresponding task $T_{T}$, transfer learning is the process of improving the target predictive function $f_{T}\left(\cdot \right)$ by using the related information from $D_{S}$ and $T_{S}$, where $D_{S}\neq D_{T}$ or $T_{S}\neq T_{T}$.

### 2.3. The Network Architecture

The network architectures resulting from the different models implemented in this study are generated according to the following stages:Input Layer;Base Model;Global Average Pooling 2D;Dropout;Dense Layer;Output Layer.

A representative diagram of the architecture used in this survey can be seen in Figure 2.

As mentioned above, firstly, the dataset is pre-processed to resize the images to 200×200 pixels. As can be seen in the representative diagram of the architecture used, the Input Layer takes pixel values of the sample that is going to be classified: that is to say, 200×200 pixels ×3 channels, where each channel corresponds to a colour of the RGB image.

Next, the processed data will enter this transfer learning model, which is the base model of the classifier. Each Keras application expects a specific type of input pre-processing, so these values will be normalised according to the base model that is selected. It should be noted that in our survey, all base models have been pre-trained with the ImageNet database [30]. That is, once the ImageNet database has been specified, the values of the weights corresponding to the base model pre-trained with this database are obtained. ImageNet is a large-scale ontology of images built upon the backbone of the WordNet [31] structure.

Subsequently, Global Average Pooling 2D refers to the pooling operation that computes the average value for spatial data across multiple layers.

The Dropout Layer randomly sets input neural network units to 0 with a frequency determined by the rate at each step during training time; it helps prevent overfitting. Inputs not set to 0 are scaled up by 1/(1 − rate) such that the sum of all inputs is unchanged. In our architecture, the rate has been set to 20%.

Afterwards, the Dense Layer, often referred to as the fully connected layer, consists of neurons connected to every neuron in the preceding layer with a specified activation function. In this study, the *Softmax* activation function has been applied.

Finally, there is the Output Layer, which is comprised of as many neurons as there are classes. Each output neuron employs the *Softmax* activation function to provide an estimation of the probability that the processed sample belongs to the corresponding class of each neuron. In our case, with four classes of species, the architecture includes four output neurons.

## 3. Experimental Methodology

This section provides a theoretical explanation of the various methods employed to derive the results in this study.

### 3.1. k-Fold Cross-Validation Method

Concerning the dimensions of the database, it is essential to consider that a dataset consisting of 4 classes, each with 40 samples, results in a relatively small dataset when compared to other studies utilizing Keras models with datasets containing thousands of samples. To address this limitation and to ensure the robustness of the classifier, the cross-validation technique has been employed in these experiments. This approach helps validate the generated models and ensures that the results are not overly influenced by the partitioning between test and training data.

Cross-validation is a resampling technique employed to assess machine learning models on a restricted dataset of samples. This method involves iteratively calculating and averaging the evaluation metrics on various partitions to provide a more comprehensive and reliable assessment of the model’s performance.

In these experiments, the training dataset and the test dataset are grouped, respectively, five times (5-folds) so that the different groupings have the same number of samples each time but have different samples. Following this, each model is trained using the training samples. Subsequently, the test dataset is classified to obtain metrics from each generated model, facilitating evaluation based on these metric values. Lastly, the results are computed as the mean of the values of these metrics obtained across the different folds. It is relevant to note the fact that the training dataset is not exactly the same in the five groupings. Each model is generated from its own training dataset, as the training dataset significantly influences the adjustments of the model, even though all of them are based on the same Keras base model type for each experiment.

The entire database is utilized in each distribution of samples and encompasses both training and test samples. However, there are various approaches to distributing and employing the original dataset. In light of this, two types of cross-validation can be discerned: exhaustive and non-exhaustive cross-validation.

Exhaustive cross-validation involves learning and testing all possible ways to divide the original sample into a training and a validation set;Non-exhaustive cross-validation methods do not compute all possible ways of splitting the original sample.

Exhaustive cross-validation methods demand significant computational resources, especially considering the dataset dimensions in this study. Specifically, in the case of *Leave-One-Out Cross-Validation (LOOCV),* the model needs to be fitted as many times as the number of samples, making it highly time-consuming, especially with 4 classes and 40 samples per class. Therefore, the cross-validation method employed in these experiments is non-exhaustive: specifically, *k-fold cross-validation (k-fold CV).*

In *k*-fold cross-validation, the dataset is randomly partitioned into *k* groups or folds of approximately equal size. The first fold is treated as a test set, and the method is fit on the remaining *k*−1 folds. This procedure is repeated *k* times, with each iteration treating a different group of samples as the test set. This iterative process yields *k* validations of the model type, eventually culminating in the computation of the mean metrics, which are used to evaluate the model. That is to say, in *k*-fold cross-validation, *k* distinct models are obtained: each derived from different training samples and all based on the same type of Keras model.

In these experiments, the training dataset and the test dataset are grouped five times (5-fold cross-validation) so that the different groupings have the same number of samples each time but different samples. With a total of 160 samples (40 samples for each species), each *k*-fold comprises 128 training samples and 32 test samples. Furthermore, data are not shuffled before each split, ensuring that no sample from the test dataset is repeated across the five different groups.

During the model training process, the training dataset of each *k*-fold is further divided into two other datasets: the training subset, which is used to train the model at each cycle (*epoch*), and the validation subset. Validation split helps to progressively improve the model performance by fine-tuning the model after each epoch.

In these experiments, a maximum of 100 epochs and a *patience* set to 20 epochs have been defined for training each model. The objective of training is to minimize the *loss*. This metric is monitored at the end of each epoch, and the training process concludes either when the *loss* no longer decreases after 20 epochs or when 100 epochs of training have been completed. The model weights are then restored to the weights from the best epoch in the training process.

The test set provides the final metrics of the model after completing the training phase. Lastly, the results are computed as the mean of the values of these metrics obtained across the different folds.

It is crucial to note the fact that the training dataset is not identical across the five groupings, resulting in the generation of five distinct models from the same architecture. Each model is created from its respective training dataset, as the training dataset significantly influences the adjustments of the model even though all of them are based on the same base model type for each experiment.

### 3.2. Performance Metrics

To cope with the great variety of the classification models, it is necessary to use metrics or comparative schemes that allow qualitative analysis of the performance of the proposed models and to contrast their results. In other words, these metrics can be employed to evaluate the efficacy of the algorithms at classifying and identifying species in the images. The definition of these metrics is based on the confusion matrix.

#### 3.2.1. Confusion Matrix

Matrix confusion is a technique that allows evaluation of the precision of the image classification algorithms. This technique assumes that the *ground truth* information is characterized by the following properties:Each image is labelled as belonging to a certain class so that there are N reference classes, ${\left\{R_{i}\right\}}_{i=1}^{N}$;Reference classes are mutually exclusive; that is to say, a certain image has no different classes (Equation (Equation 1)):
(1)$$R_{i}\cap R_{j}=⌀,\nabla i\neq j$$

Assuming that each sample $R_{i}$ from a particular species *S* to be evaluated is assigned by an algorithm as belonging to a certain class $C_{i}$ and having *N* classes, the dataset $C_{i}$ determines only one specific species to evaluate, meaning that two different sets have no elements in common. Ultimately, there is no more than one species of the four classes under study in each image in these experiments. This can be expressed mathematically as indicated in Equation (Equation 2).
(2)$$\cup_{i=1}^{N}C_{i}\in S\;\mathrm{and}\;C_{i}\cap C_{j}=⌀,\nabla i\neq j$$

A binary classifier model can be established in which the results are tagged as positives (*p*) or negatives (*n*). In this theoretical framework, the prediction issue offers four possible results from the classification carried out, where:*TP* is true positive: a test result that correctly indicates the presence of a condition or characteristic;*TN* is true negative: a test result that correctly indicates the absence of a condition or characteristic;*FP* is false positive: a test result that wrongly indicates that a particular condition or attribute is present;*FN* is false negative: a test result that wrongly indicates that a particular condition or attribute is absent.

Based on the above, an experiment can be defined with *P* positive instances and *N* negative instances. The four possible outcomes can be represented in a 2×2 confusion matrix (Table 2).

From this confusion matrix, various metrics can be derived to evaluate the performance of different prediction models. The performance of the classification algorithms from this research was mainly evaluated using four metrics: *Accuracy*, *Precision*, *Recall* and *F1 Score*.

#### 3.2.2. *Accuracy*

The *Accuracy* is defined as the fraction of correct predictions made by the classifier out of the total number of predictions. *Accuracy* can also be calculated in terms of positive and negative predictions as expressed by Equation (Equation 3):(3)$$Accuracy=\frac{TP+TN}{TP+TN+FP+FN}$$

#### 3.2.3. *Precision*

The *Precision*, also called *Positive Predictive Value (PPV)*, is the fraction of test images classified as a specific class—as an example, class A—that are truly assigned to this class. *Precision* can be calculated as expressed by Equation (Equation 4):(4)$$Precision=\frac{TP}{TP+FP}$$

#### 3.2.4. *Recall*

*Recall*, also known as *Sensitivity, Hit Rate* or *True Positive Rate (TPR),* is the fraction of test images from a class that are correctly identified to be assigned to this class. *Recall* can be calculated as expressed by Equation (Equation 5):(5)$$Recall=\frac{TP}{TP+FN}$$

#### 3.2.5. *F1 Score*

The last two metrics can be used as parts of another metric that gives the average of the *Precision* and *Recall*. This can be interpreted as the *F1 Score*, for which the best value is 1 and the worst value is 0. The *F1 Score* can be calculated as expressed by Equation (Equation 6):(6)$$F1\;Score=2\cdot \left(\frac{Precision\cdot Recall}{Precision+Recall}\right)$$

## 4. Results

This section presents the outcomes obtained from the classification experiments conducted with the implemented models. Based on these results, various aspects of the comparison are discussed.

As mentioned earlier, each training session is conducted with a maximum of 100 epochs for each *k*-fold. However, some models completed training in a lower number of epochs across the 5-folds. Additionally, since the base models were configured in a manner such that their internal parameters were not altered during training, their respective weights and biases remain constant. Hence, there are both non-trainable parameters belonging to the base model in use and trainable parameters belonging to the rest of the neural network in each model.

Table 3 displays the maximum number of epochs for training the models and the parameter count for each model. The first column includes row identifiers for ease of reading, while the second column lists the names of the base models. The third column indicates the maximum number of epochs, which corresponds to the *k*-fold with the most epochs. The fourth column shows the total number of parameters in the entire network, and the last column specifies the count of trainable parameters.

Similarly, Table 4 shows the metrics obtained—*Accuracy*, *Precision*, *Recall* and *F1 Score*—depending on the base model integrated in each model. Since the experiments involved a cross-validation 5-fold method, these values are actually the means in percentage from all *k*-folds for each metric and their corresponding standard deviations.

Considering the results presented in both tables, it can be observed that in general, models with a higher maximum number of epochs demonstrate better performance, as they had more opportunities to learn.

The architecture with the highest number of total parameters is the one implemented with the base model *EfficientNetV2L.* As can be observed, this model has a total of 117,751,972 parameters, of which only 5124 are trainable. This suggests that even though the base model has a very high number of neurons in its hidden layers, there are not many units in its last hidden layer if the architecture is compared with another one, such as the one that implements the *EfficientNetB7* base model. The last one has almost half of the total parameters: 64,107,931; however, 10,244 of them are trainable. It is the model that has the highest number of trainable parameters.

The models with the lowest number of trainable parameters, 2052, are those that implement the base models *VGG16* or *VGG19*. The architecture with the lowest number of total parameters, 2,263,108, is the one with the base model *MobileNetV2.* Regarding the metrics, the model with *MobileNetV2* does not provide favourable results.

Nevertheless, despite the fact that the models with *VGG16* and *VGG19* have the lowest number of trainable parameters, their metrics are quite favourable—around 80%—compared to others such as the model with *InceptionV3,* which has more total and trainable parameters but for which its metrics have disadvantageous values.

From this comparison, the model with the highest metrics is the one with *EfficientNetV2B3,* for which the values exceed 98%, and both the total and trainable parameters for this model are not as high as for other models. That is to say, even though others have used a larger number of parameters, they have not been able to achieve the performance offered by *EfficientNetV2B3*. The values of the different metrics and the number of parameters obtained through this model, drawn, respectively, from Table 3 and Table 4, are summarised as follows:Total parameters: **12,936,770;**Trainable parameters: **6148**;*Accuracy* - Mean: **98.75%**;*Accuracy* - Standard Deviation: **1.53**;*Precision* - Mean: **98.60%**;*Precision* - Standard Deviation: **1.95**;*Recall* - Mean: **98.80%**;*Recall* - Standard Deviation: **1.79**;*F1 Score* - Mean: **98.60%**;*F1 Score* - Standard Deviation: **1.95**.

Considering the total number of samples in the database, there are 32 samples to be classified in each *k*-fold. So it must be said that only two classification errors, each occurring in different *k*-folds, have resulted in these metrics for this model.

## 5. Discussion

This research has served as a significant starting point for the automatic identification of invasive alien species and endemic species in the Canary Islands. It has demonstrated the potential of implementing transfer learning models as a part of neural network models, where one of the most remarkable aspects is the number of models tested. The comparison includes 31 models implemented from different Keras base models.

Based on the outcomes of the experiments conducted in this research, it can be stated that while certain models implemented with Keras exhibit low-performance classification, others represent a promising approach for the automated identification of these specific species, which are relevant to the preservation of the fauna of this archipelago.

In addition, the research conducted in this study has demonstrated that certain implemented base models exhibit a more favourable trend in the classification of the species under study. Consequently, these models could be specifically considered in the development of a practical system for identifying these particular species. Notably, they have shown promising results even when subjected to samples from the *Wild Dataset*, implying successful performance under adverse visibility conditions for the species. Although the images in this study were sourced from various internet platforms, the insights gained could be applied in future experiments using images captured with camera traps, given their similar visibility characteristics. Thus, the findings of this research hold promise for the development of monitoring systems based on camera traps for real-world applications.

In the comparison, the model that stands out among all others is the one implementing the base mode *EfficientNetV2B3.* This particular model has demonstrated superior performance by achieving the best outcomes for all metrics and incurring only two classification errors.

Beforehand, it could be thought that due to chance, the samples were grouped in the training, evaluation, and test sets in a manner that led to overly positive results. In other words, samples causing higher classification errors could have been included in the training set, while those leading to fewer errors were used in the test set.

Certainly, chance does play a fundamental role in sample distribution and influences the results. However, it is important to note that the 5-fold cross-validation methodology was employed, and the samples were not shuffled before creating each fold in this study. Consequently, each sample was part of the test set in one of the folds. This methodology is widely used in numerous publications and is considered a standard in model validation; this illustrates the variability of cases regarding whether or not a sample belongs to the test set.

Taking this into consideration, the primary variable in this study was the base model itself, and each model exhibited distinct performance characteristics due to the uniqueness of the data. A considerable number of models underwent testing, and based on the results obtained, it is possible to categorize them into groups according to their performance. Notably, the base models that yielded the best results belong to the *EfficientNet* family: achieving a mean *Accuracy* of 90% and above. Following closely, some of the *ResNet* models produced results around the 90% mark. Subsequently, both the *VGG16* and *VGG19* models surpassed 80%. Finally, the remaining models demonstrated a substantial decrease in efficiency.

In conclusion, it is essential to highlight that even the models delivering the best results had limitations in their learning stages due to the maximum value of epochs during training. Hence, adjusting this hyperparameter could be explored in future research. Furthermore, with regard to the use of the most favourable base models, potential improvements in the architecture could be considered for subsequent work. For instance, incorporating an Attention Layer or Transformers or applying Ensemble Learning techniques might be worth exploring. Lastly, given the relatively low number of samples in the database and the notable physical differences between the species, the optimistic results obtained by these models should be interpreted with caution. Therefore, future studies should increase the number of classes and, more importantly, the number of samples per class to ensure that the research yields results that are conducive to the development phase. 

## Figures and Tables

**Figure 1 sensors-24-01372-f001:**
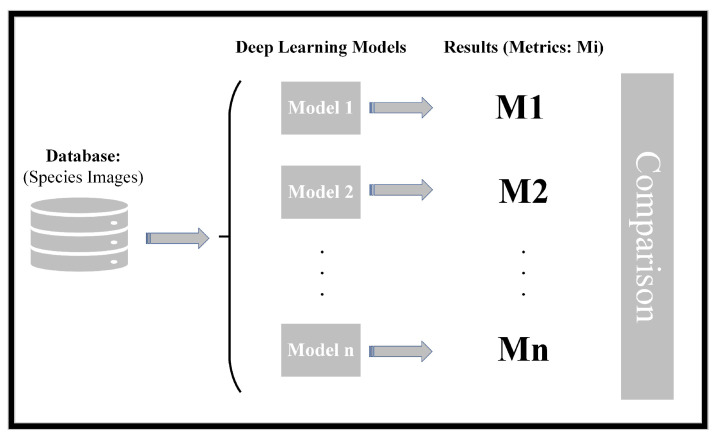
Conceptual schematic representation of the work carried out.

**Figure 2 sensors-24-01372-f002:**
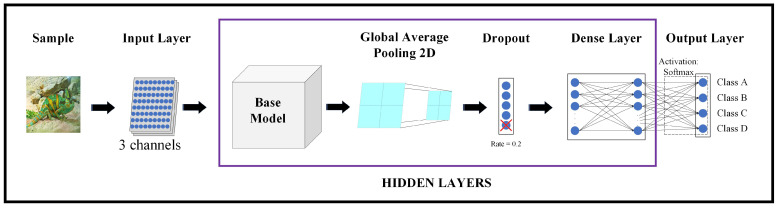
Representative diagram of the architecture.

**Table 1 sensors-24-01372-t001:** Dataset of the selected samples.

Class	Number of Samples	Colour Model	Format	Aspect Ratio (Pixels)
*G. stehlini*	40	RGB	JPEG	200×200
*C. sexlineatus*	40	RGB	JPEG	200×200
*C. calyptratus*	40	RGB	JPEG	200×200
*P. regius*	40	RGB	JPEG	200×200

**Table 2 sensors-24-01372-t002:** Confusion matrix (2×2).

		PREDICTION	
		Positive Prediction	Negative Prediction
**GROUND-TRUTH** **CONDITION**	**Positive Condition**	True Positives (TP)	False Negatives (FN)
	**Negative Condition**	False Positives (FP)	True Negatives (TN)

**Table 3 sensors-24-01372-t003:** Maximum number of epochs and number of parameters in the implemented models.

ID	Base Model	No. Epochs (Maximum)	Parameters (Weights + Biases)	
			Total	Trainable
1	Xception	33	20,869,676	8196
2	VGG16	63	14,716,740	2052
3	VGG19	86	20,026,436	2052
4	ResNet50	99	23,595,908	8196
5	ResNet50V2	37	23,572,996	8196
6	ResNet101	100	42,666,372	8196
7	ResNet101V2	56	42,666,372	8196
8	ResNet152	88	58,379,140	8196
9	ResNet152V2	37	58,339,844	8196
10	InceptionV3	22	21,810,980	8196
11	InceptionResNetV2	41	54,342,884	6148
12	MobileNet	69	3,232,964	4100
13	MobileNetV2	62	2,263,108	5124
14	DenseNet121	75	7,041,604	4100
15	DenseNet169	65	12,649,540	6660
16	DenseNet201	75	18,329,668	7684
17	EfficientNetB0	100	4,054,695	5124
18	EfficientNetB1	100	6,580,363	5124
19	EfficientNetB2	100	7,774,205	5636
20	EfficientNetB3	100	10,789,683	6148
21	EfficientNetB4	100	17,680,995	7172
22	EfficientNetB5	100	28,521,723	8196
23	EfficientNetB6	100	40,969,363	9220
24	EfficientNetB7	100	64,107,931	10,244
25	EfficientNetV2B0	100	5,924,436	5124
26	EfficientNetV2B1	100	6,936,248	5124
27	EfficientNetV2B2	100	8,775,010	5636
28	EfficientNetV2B3	100	12,936,770	6148
29	EfficientNetV2S	100	20,336,484	5124
30	EfficientNetV2M	100	53,155,512	5124
31	EfficientNetV2L	100	117,751,972	5124

**Table 4 sensors-24-01372-t004:** Values of the metrics in the implemented models.

ID	Base Model	*Accuracy*	*Precision*	*Recall*	*F1 Score*
		Mean (%) (Standard Deviation)			
1	Xception	37.50 (6.85)	34.40 (7.80)	37.40 (7.16)	34.00 (7.69)
2	VGG16	81.25 (9.06)	81.60 (9.29)	82.20 (9.88)	80.00 (11.53)
3	VGG19	81.25 (7.13)	81.20 (8.93)	83.20 (9.88)	79.80 (11.53)
4	ResNet50	93.75 (4.42)	94.80 (4.32)	93.60 (5.32)	93.80 (5.07)
5	ResNet50V2	32.50 (7.02)	32.80 (10.18)	36.00 (8.37)	28.60 (5.55)
6	ResNet101	92.50 (7.02)	91.80 (8.50)	92.60 (6.84)	91.40 (8.59)
7	ResNet101V2	90.00 (3.06)	89.60 (3.78)	90.40 (2.88)	89.20 (2.86)
8	ResNet152	89.37 (2.50)	89.60 (3.21)	89.40 (2.70)	89.20 (3.56)
9	ResNet152V2	31.87 (7.23)	27.40 (4.93)	31.60 (8.90)	26.00 (7.18)
10	InceptionV3	29.37 (8.05)	33.40 (12.60)	30.40 (8.79)	28.40 (9.07)
11	InceptionResNetV2	30.62 (8.24)	33.40 (12.58)	31.40 (9.71)	25.40 (8.26)
12	MobileNet	58.12 (8.29)	59.40 (10.62)	59.20 (8.87)	57.40 (10.78)
13	MobileNetV2	48.75 (6.12)	52.20 (3.35)	50.20 (6.87)	49.00 (6.63)
14	DenseNet121	55.62 (11.07)	59.80 (13.42)	55.40 (12.44)	53.20 (12.75)
15	DenseNet169	58.75 (8.24)	56.60 (9.42)	57.00 (9.19)	56.20 (9.28)
16	DenseNet201	54.37 (15.00)	58.20 (16.11)	55.20 (16.15)	54.40 (16.38)
17	EfficientNetB0	96.87 (1.98)	96.40 (2.51)	97.40 (2.19)	96.60 (2.51)
18	EfficientNetB1	98.12 (1.53)	97.60 (2.30)	98.20 (1.79)	97.80 (2.05)
19	EfficientNetB2	96.87 (1.97)	96.00 (4.12)	97.40 (1.82)	96.40 (3.29)
20	EfficientNetB3	97.50 (2.34)	97.40 (2.88)	97.60 (2.30)	97.40 (2.88)
21	EfficientNetB4	96.87 (1.98)	96.40 (2.30)	97.00 (2.55)	96.80 (2.17)
22	EfficientNetB5	95.00 (2.50)	95.20 (2.68)	95.40 (2.88)	95.20 (2.68)
23	EfficientNetB6	95.62 (3.19)	96.00 (3.81)	95.60 (3.21)	95.40 (3.91)
24	EfficientNetB7	95.00 (1.53)	95.20 (1.64)	94.40 (2.07)	94.60 (1.95)
25	EfficientNetV2B0	97.50 (2.34)	97.80 (2.49)	98.00 (2.12)	97.80 (2.49)
26	EfficientNetV2B1	98.12 (1.53)	98.20 (1.79)	97.60 (2.51)	97.80 (2.17)
27	EfficientNetV2B2	97.50 (3.64)	97.80 (3.49)	98.20 (3.03)	97.80 (3.49)
28	EfficientNetV2B3	**98.75 (1.53) ^1^**	**98.60 (1.95) ^1^**	**98.80 (1.79) ^1^**	**98.60 (1.95) ^1^**
29	EfficientNetV2S	97.50 (2.34)	97.80 (2.49)	97.80 (2.17)	97.60 (2.51)
30	EfficientNetV2M	96.87 (3.42)	97.00 (3.67)	97.20 (3.27)	97.00 (3.67)
31	EfficientNetV2L	97.50 (3.64)	97.20 (5.21)	97.20 (4.38)	97.20 (4.76)

^1^ Best outcomes.

## Data Availability

The data presented in this study are available on request from the corresponding author. The database is not publicly available due to some data being copyrighted.

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
