# Peer review of "Reptile Identification for Endemic and Invasive Alien Species Using Transfer Learning Approaches"

_sensors, 2024, doi:10.3390/s24051372_

Round 1

Reviewer 1 Report

Comments and Suggestions for Authors

The manuscript addresses a significant ecological issue in the Canary Islands concerning the identification of endemic and invasive reptile species. The study proposes the use of deep learning techniques, specifically transfer learning, to automatically identify certain species of reptiles. The research focuses on implementing different neural network models, culminating in the selection of EfficientNetV2B3 as the base model with a reported mean accuracy of 98.75%. The manuscript represents a valuable contribution to the field of species identification and ecological management. Some suggestions are given below:

Provide a more detailed background on the ecological impact of invasive alien species in the Canary Islands. Clearly state the significance of the research problem and how an automated identification system can aid in the conservation efforts initiated by regional authorities. Besides, introduce more relevant work on the application of deep learning and transfer learning methods, such as: doi.org/10.1007/s00170-022-10335-8, and doi.org/10.3390/s22249926.

Clarify the dataset used for training and testing the models. Include information on the number of species, the size of the dataset, and any data augmentation techniques employed. Include a more in-depth discussion of the results, particularly the factors contributing to the high mean accuracy of 98.75%. Discuss challenges encountered during the training process and how they were addressed.

Consider discussing any limitations or challenges encountered during the study and propose directions for future research to address these limitations.

Overall, the manuscript addresses an important ecological issue and presents a promising approach using transfer learning for reptile identification. Addressing the specific comments and suggestions will enhance the clarity and impact of the research.

Comments on the Quality of English Language

The language and style of the manuscript are generally clear and concise, effectively conveying the key ideas and findings.

Author Response

**********************************************************************
Comment 1
Provide a more detailed background on the ecological impact of invasive alien species in the Canary Islands.
---------------------------------------------------------------------------------------------------------
Answer to Comment 1
Thank you for the useful comments.
The authors have added the following lines in the third paragraph of section 1, to provide a more detailed background on the ecological impact of invasive alien species in the Canary Islands:
“Furthermore, the vulnerability to invasions is significantly heightened in the Canary Islands, attributed to the distinctive ecological conditions under which island organisms have evolved. In other words, the absence of adaptations to predators, low genetic diversity, and increased susceptibility to exotic pathogens, among other factors, amplify the detrimental effects of biological invasions in the Canary Islands compared to continental ecosystems[5].”
**********************************************************************
Comment 2
Clearly state the significance of the research problem and how an automated identification system can aid in the conservation efforts initiated by regional authorities.
---------------------------------------------------------------------------------------------------------
Answer to Comment 2
The authors have added the following sixth paragraph in Section 1, to state the significance of the research problem and how an automated identification system can aid in the conservation efforts initiated by regional authorities:
“To address the challenges posed by the absence of sightings or reported presence of relevant species, we propose the implementation of an automatic monitoring system. This system would utilize volumetric motion sensors and camera traps to detect and record the presence of species more proactively and continuously. Volumetric motion sensors can be employed to activate camera traps upon detecting the presence of species in pre-defined spaces. The integration of these motion sensors with camera traps allows for the automated recording of images when triggered by the detected motion. This particular camera type provides the capability for continuous real-time monitoring. By incorporating automatic identification algorithms, the system can promptly activate an alert signal, notifying system administrators of the presence of any of the relevant species. The identification of species through strategically placed
cameras in the Canary Islands will empower regional authorities to implement targeted measures. This includes actions like capturing the identified specimens and providing improved ecosystem monitoring.”
**********************************************************************
Comment 3
Besides, introduce more relevant work on the application of deep learning and transfer learning methods, such as: doi.org/10.1007/s00170-022-10335-8, and doi.org/10.3390/s22249926.
---------------------------------------------------------------------------------------------------------
Answer to Comment 3
The authors have added the following paragraph with its corresponding references as the penultimate paragraph of the section 1.1, to add more relevant work on the application of deep learning and transfer learning methods.
“Furthermore, the Transfer Learning method can be effectively combined with some of the aforementioned techniques. Indeed, Transfer Learning has found application in various works across diverse domains. In the realm of wildlife identification, this method has been utilized for fish identification in tropical waters [13], distinguishing between different dog breeds [14] or accurately identifying various bird species [15].”
**********************************************************************
Comment 4
Clarify the dataset used for training and testing the models. Include information on the number of species, the size of the dataset, and any data augmentation techniques employed.
---------------------------------------------------------------------------------------------------------
Answer to Comment 4
The authors have replaced the two last paragraphs of section 3.1 with the following five paragraphs, to clarify the dataset used for training and testing the models. In addition, information on the number of species and the size of the dataset has been specified. Data augmentation techniques have not been employed so no information in this regard has been reflected in the text.
“In these experiments, the training dataset and the test dataset are grouped five times (5-folds), so that the different groupings have the same number of samples each time but different samples. With a total of 160 samples (40 samples for each species), each k-fold comprises 128 training samples and 32 test samples. Furthermore, data are not shuffled before each split, ensuring that no sample from the test dataset is repeated across the five different groups.
During the model training process, the training dataset of each 5-fold is further divided into two other data sets: the training subset, which is used to train the model at each cycle (epoch), and the validation subset. Validation split helps to progressively improve the model performance by fine-tuning the model after each epoch.
In these experiments, a maximum of 100 epoch and a patience set to 20 epoch have been defined for training each model. The objective of training is to minimize the loss. This metric is monitored at the end of each epoch and the training process concludes either when the loss no longer decreases after 20 epochs or when 100 epochs of training have been completed. The model weights are then restored to the weights from the best epoch in the training process.
The test set provides the final metrics of the model after completing the training phase. Lastly, the results are computed as the mean of the values of these metrics obtained across the different folds.
It is crucial to note the fact that the training dataset is not identical across the five groupings, resulting in the generation of 5 distinct models from the same architecture. Each model is created from its respective training dataset, as the training dataset significantly influences the adjustments of the model. Even though all of them are based on the same base model type for each experiment.”
**********************************************************************
Comment 5
Include a more in-depth discussion of the results, particularly the factors contributing to the high mean accuracy of 98.75%. Discuss challenges encountered during the training process and how they were addressed.
---------------------------------------------------------------------------------------------------------
Answer to Comment 5
The authors have added the following paragraphs (the fifth, sixth and seventh paragraph) in section 5 to include a more in-depth discussion of the results:
“Beforehand, it could be thought that, due to chance, the samples were grouped in the training, evaluation, and test sets in a manner that led to overly positive results. In other words, samples causing higher classification errors could have been included in the training set, while those leading to fewer errors were used in the test set.
Certainly, chance does play a fundamental role in sample distribution, influencing the results. However, it is important to note that the 5-fold Cross-Validation methodology has been employed, and the samples are not shuffled before creating each fold in this study. Consequently, each sample is part of the test set in one of the folds. This methodology is widely used in numerous publications and is considered a standard in model validation, illustrating the variability of cases regarding whether or not a sample belongs to the test set.
Taking this into consideration, the primary variable in this study has been the base model itself, and each model exhibits distinct performance characteristics due to the uniqueness of the data. A considerable number of models have undergone testing, and based on the results obtained, it is possible to categorize them into groups according to their performance. Notably, the base models that yielded the best results belong to the EfficientNet family, achieving a mean accuracy of 90% and above. Following closely, some of the ResNet models produced results around the 90% mark. Subsequently, both the VGG16 and VGG19 models surpassed 80%. Finally, the remaining models demonstrated a substantial decrease in efficiency.”
**********************************************************************
Comment 6
Consider discussing any limitations or challenges encountered during the study and propose directions for future research to address these limitations.
---------------------------------------------------------------------------------------------------------
Answer to Comment 6
In addition to the aspects discussed in the previous paragraphs, the authors have changed the last paragraph in section 5 to clarify the limitation of epoch hyperparameter, so, we propose to increment the maximum number of epochs and other suggestions in future line research.
This is the last paragraph:
“In conclusion, it is essential to highlight that even the models delivering the best results had limitations in their learning stages due to the maximum value of epochs during training. Hence, adjusting this hyperparameter could be explored in future research. Furthermore, with regard to the use of the most favourable base models, potential improvements in architecture could be considered for subsequent work. For instance, incorporating an Attention Layer, Transformers or applying Ensemble Learning techniques might be worth exploring. Lastly, given the relatively low number of samples in the database and the notable physical differences between the species, the optimistic results obtained by these models should be interpreted with caution. Therefore, future studies should increase the number of classes and, more importantly, the number of samples per class to ensure that the research yields results that are conducive to the development phase.”
**********************************************************************
Comment 7
Comments on the Quality of English Language
The language and style of the manuscript are generally clear and concise, effectively conveying the key ideas and findings.
---------------------------------------------------------------------------------------------------------
Answer to Comment 7
Thanks a lot for your comment.
As a note, the authors have improved the wording.
**********************************************************************

Reviewer 2 Report

Comments and Suggestions for Authors

The manuscript presents an interesting approach to automated species identification and has the potential to be very useful to a wide range of researchers. However, I think that there are some methodological issues that need to be clarified before publication. I am particularly concerned about the small sample size of the authors and the vague explanations for the existing base models that were used. I have no experience with these models for images, but I have used some of them for sound analyses (i.e., identifying species by its sound), and in such it is important to include records where the focus species is missing, so the algorithm may train correctly. The lower the number of records for training, the greater the chance a similar sound will be erroneously identified as the sound of the focus species. I think that the high accuracy of the image models might be due to the fact that the four study species have very different overall appearance, and in real-life scenarios, when analysed images will be of multitude of species – some of them very similar to each other - the results might be vastly different. The process behind model building needs to be explained better – at present, while there is plenty of theoretical explanations in the Methods section, the practical side of what the authors have actually used remains somewhat unclear. I still think that the model could be very useful for highlighting alien species that look very different to local fauna (or other unique-looking species in a given environment), but the authors need to better explain and defend its usefulness for general species identification. Below are my in-text comments:

Lines 144-150: Only genus names should be capitalised (i.e., Gallotia stehlini). I would also suggest to include the Author name for each of the study species when they are first mentioned - i.e., the species is first introduced as the Gran Canaria giant lizard, Gallotia stehlini (Schenkel, 1901) and subsequently referred to as G. stehlini or by its vernacular name.

Figure 2: Again, correct the Latin names.

Figure 3: I think that figures 2 and 3 with their accompanying text should be in Appendix, not in the main text of the manuscript.

Line 209: Isn't 40 too low a number? And from this it appears that you have not included images that contain neither of the study species?

Lines 217-233: My concern is that you appear to compare four very different-looking species, and that is the main reason for your high accuracy. Under real-world circumstances, the automated recognition will have to differentiate between similar looking species, some of which will be your study species and others will belong to different species.

Lines 272-273: I would like to see more information on those pre-trained models with millions of images from the internet.  What exactly have you used?

Figure 4: There is a spelling error in Figure 4 - "Poling" instead of "Pooling".

Lines 308-309: How was that done? What is this database? Provide more information on this.

Lines 372-373: I remain unconvinced in the validity of your approach after this explanation. Why haven't you included more species in your testing, what exactly are these base models and what are the databases they use?

Tables 3 and 4: I suggest that tables 3 and 4 are combined in a single Table in a landscape layout. Currently they just present different parameters for the same models.

Lines 487-491: The automatic identification would probably be useful for the two invasive species, which are rather different to every local species, but I doubt you will have the same accuracy when comparing similar-looking local species.

Lines 492-495: It is unclear why some models were better than others. Is it possible that with other species, results may vary drastically, or are your models independent of the study species? Provide more information in this regard.

Line 497: What do you mean with "this type of species"?

Lines 498-500: I think this is a really speculative statement that is not backed by your actual results - your photos were not from camera traps.

Comments on the Quality of English Language

At times the wording is awkward and the style is just strange (e.g., "The study culminates with a comparison...", "...all of these are reptiles which are interesting to ecological concerns.", etc.). I think that the manuscript should be checked by a proficient English speaker.

Author Response

**********************************************************************
Comment 1
Lines 144-150: Only genus names should be capitalised (i.e., Gallotia stehlini). I would also suggest to include the Author name for each of the study species when they are first mentioned - i.e., the species is first introduced as the Gran Canaria giant lizard, Gallotia stehlini (Schenkel, 1901) and subsequently referred to as G. stehlini or by its vernacular name.
---------------------------------------------------------------------------------------------------------
Answer to Comment 1
Thank you for the useful comments.
The authors have included the Author name for each of the study species when they are first mentioned (third paragraph of section 2.1), and subsequently referred to as you suggested. The paragraph in which the species names are mentioned for the first time has been reworded as follows:
“The selected reptile species included in the database are of ecological significance, either inhabiting or having been observed on Gran Canaria island. Specifically, two of these species are documented as having particular ecological importance and are registered as such in the State Catalogue [18]: both the Gran Canaria giant lizard, Gallotia stehlini (Schenkel, 1901) and the Gran Canaria skink, Chalcides sexlineatus (Steindachner, 1891). The remaining two species in the dataset are the Yemen chameleon, Chamaeleo calyptratus (Duméril & Duméril, 1851) and the ball python, Python regius (Shaw, 1802). Both of these species are categorized as invasive alien species of concern for the outermost region of the Canary Islands, as outlined in the Spanish Catalogue of invasive alien species [19].”
**********************************************************************
Comment 2
Figure 2: Again, correct the Latin names.
---------------------------------------------------------------------------------------------------------
Answer to Comment 2
The authors have corrected the Latin names of the Figure 2 and Figure 3, using the genus name capitalized.
**********************************************************************
Comment 3
Figure 3: I think that figures 2 and 3 with their accompanying text should be in Appendix, not in the main text of the manuscript.
---------------------------------------------------------------------------------------------------------
Answer to Comment 3
The authors have moved the figure 2 and figure 3 to the Appendix A.1 and Appendix A.2 respectively.
**********************************************************************
Comment 4
Line 209: Isn't 40 too low a number? And from this it appears that you have not included images that contain neither of the study species?
---------------------------------------------------------------------------------------------------------
Answer to Comment 4
The answer to your question is yes, undoubtedly, the authors of this study are aware that 40 samples is too low a number. The reason for using a dataset with so few samples is because the samples have been selected one at a time from internet searches.
In general, the images available on specialised websites usually present the photographed specimens in very favourable visibility conditions. However, finding images on the internet of species photographed in unfavourable conditions for identification (Wild Dataset) is a complicated task due to the scarcity of such images, especially for endemic species in the Canary Islands.
Nevertheless, despite the low number of samples available in this study, authors have considered it sufficient for this initial approach.
On the other hand, the authors have changed this line with the following text to clarify this dataset corresponds to the species under study.
“For the experiments in this research, a balanced dataset of the species under study was employed, with 40 samples per class”
**********************************************************************
Comment 5
Lines 217-233: My concern is that you appear to compare four very different-looking species, and that is the main reason for your high accuracy. Under real-world circumstances, the automated recognition will have to differentiate between similar looking species, some of which will be your study species and others will belong to different species.
---------------------------------------------------------------------------------------------------------
Answer to Comment 5
The authors agree with you that if the species under study were very similar to each other, the accuracy values would most likely not be so high. However, the comparison made in this initial approach is useful to know which base models offer better performance and which ones offer poor performance, even though these species are very different from each other.
That is to say, InceptionV3 is the base model from which the lowest mean accuracy value is obtained in this study (29.37%). This information suggests that InceptionV3 can be discarded when designing a system that can be used in a real application, while the base models that offer better performance should be taken into account when designing more capable systems.
The authors have included the following last 2 paragraphs in section 5 clarifying this issue:
“Taking this into consideration, the primary variable in this study has been the base model itself, and each model exhibits distinct performance characteristics due to the uniqueness of the data. A considerable number of models have undergone testing, and based on the results obtained, it is possible to categorize them into groups according to their performance. Notably, the base models that yielded the best results belong to the EfficientNet family, achieving a mean accuracy of 90% and above. Following closely, some of the ResNet models produced results around the 90% mark. Subsequently, both the VGG16 and VGG19 models surpassed 80%. Finally, the remaining models demonstrated a substantial decrease in efficiency.
In conclusion, it is essential to highlight that even the models delivering the best results had limitations in their learning stages due to the maximum value of epochs during training. Hence, adjusting this hyperparameter could be explored in future research. Furthermore, with regard to the use of the most favourable base models, potential improvements in architecture could be considered for subsequent work. For instance, incorporating an Attention Layer, Transformers or applying Ensemble Learning techniques might be worth exploring. Lastly, given the relatively low number of samples in the database and the notable physical differences between the species, the optimistic results obtained by these models should be interpreted with caution. Therefore, future studies should increase the number of classes and, more importantly, the number of samples per class to ensure that the research yields results that are conducive to the development phase.”
**********************************************************************
Comment 6
Lines 272-273: I would like to see more information on those pre-trained models with millions of images from the internet. What exactly have you used?
---------------------------------------------------------------------------------------------------------
Answer to Comment 6
It must be noted that the base models can be adjusted with different parameters. It could use a different activation function, a pooling mode for feature extraction… or even a random initialization of their weights. In our case, the Transfer Learning technique has been applied, so we have used the pre-trained base models with ImageNet dataset, such as it has been specified in section 2.3 (fifth paragraph):
“in our survey, all base models have been pre-trained with the ImageNet database [30]”.
Nevertheless, as can be seen in the diagram of the architecture, the pre-trained base model is only a part of the complete model. The complete model fits the 4 classes under study. It is trained, validated and tested in each folder.
Each of the base models used does not vary in the training stage (this part of the architecture remains "frozen"), however, the other components of the neural network that make up the complete model do adjust in each training stage according to the 4 classes under study. And this adjustment of the neural network is done 5 times (5-fold) with each of the 31 base models.
**********************************************************************
Comment 7
Figure 4: There is a spelling error in Figure 4 - "Poling" instead of "Pooling".
---------------------------------------------------------------------------------------------------------
Answer to Comment 7
Thank you for this warning, the authors have corrected this spelling error in the new version of the paper.
**********************************************************************
Comment 8
Lines 308-309: How was that done? What is this database? Provide more information on this.
---------------------------------------------------------------------------------------------------------
Answer to Comment 8
After these lines, the authors have added the following lines (section 2.3, fifth paragraph) to clarify these questions:
“That is, once the ImageNet database has been specified, the values of the weights corresponding to the base model pre-trained with this database are obtained. ImageNet
is a large-scale ontology of images built upon the backbone of the WordNet [31] structure.”
**********************************************************************
Comment 9
Lines 372-373: I remain unconvinced in the validity of your approach after this explanation. Why haven't you included more species in your testing, what exactly are these base models and what are the databases they use?
---------------------------------------------------------------------------------------------------------
Answer to Comment 9
The authors have replaced the two last paragraphs of section 3.1 with the following five paragraphs, to clarify the dataset used for training and testing the models.
“In these experiments, the training dataset and the test dataset are grouped five times (5-folds), so that the different groupings have the same number of samples each time but different samples. With a total of 160 samples (40 samples for each species), each k-fold comprises 128 training samples and 32 test samples. Furthermore, data are not shuffled before each split, ensuring that no sample from the test dataset is repeated across the five different groups.
During the model training process, the training dataset of each 5-fold is further divided into two other data sets: the training subset, which is used to train the model at each cycle (epoch), and the validation subset. Validation split helps to progressively improve the model performance by fine-tuning the model after each epoch.
In these experiments, a maximum of 100 epoch and a patience set to 20 epoch have been defined for training each model. The objective of training is to minimize the loss. This metric is monitored at the end of each epoch and the training process concludes either when the loss no longer decreases after 20 epochs or when 100 epochs of training have been completed. The model weights are then restored to the weights from the best epoch in the training process.
The test set provides the final metrics of the model after completing the training phase. Lastly, the results are computed as the mean of the values of these metrics obtained across the different folds.
It is crucial to note the fact that the training dataset is not identical across the five groupings, resulting in the generation of 5 distinct models from the same architecture. Each model is created from its respective training dataset, as the training dataset significantly influences the adjustments of the model. Even though all of them are based on the same base model type for each experiment.”
In addition, as we have commented in our answer to your “Comment 5”, this initial approach is useful to know which base models offer better performance. This is the first step in the research, but more species will be progressively included. These species have
been selected in the first place because, at present, invasive reptile species are proliferating in Gran Canaria, and putting at risk the endemic reptiles of this island, such as the species studied.
The base models are Convolutional Neural Networks which form part of the proposed network architecture, and which have been previously trained with the images from ImageNet database.
**********************************************************************
Comment 10
Tables 3 and 4: I suggest that tables 3 and 4 are combined in a single Table in a landscape layout. Currently they just present different parameters for the same models.
---------------------------------------------------------------------------------------------------------
Answer to Comment 10
Thank you very much for this suggestion. Such as you have noted both tables present the same models and all their parameters are shared by these models so that both tables can be merged. However, the authors have considered presenting the data in these two tables for readability reasons.
If only one table is used to display all the data of the 31 models, the data density is very high, therefore it has been chosen to display the data by distributing the different parameters in these 2 tables.
**********************************************************************
Comment 11
Lines 487-491: The automatic identification would probably be useful for the two invasive species, which are rather different to every local species, but I doubt you will have the same accuracy when comparing similar-looking local species.
---------------------------------------------------------------------------------------------------------
Answer to Comment 11
Of the four species under study, two of these species are invasive species, while the other two, somewhat more similar to each other, are local species. The authors have added to the discussion (Section 5) the last 4 paragraphs explaining in more depth how the C-V 5-fold method can tackle the casuistry of the images used to find out which models offer the best performance, and based on these models, in the last paragraph, changes in the architecture are proposed to deal with a database with a higher number of species.
**********************************************************************
Comment 12
Lines 492-495: It is unclear why some models were better than others. Is it possible that with other species, results may vary drastically, or are your models independent of the study species? Provide more information in this regard.
---------------------------------------------------------------------------------------------------------
Answer to Comment 12
The authors think it depends on the casuistry of the type of image and the search for differences between these species. If the differences were clearer, by color and other characteristics, or differentiating elements, perhaps the models with the best results would have been different, so we tested 31 models, to obtain the best model based on heuristic experimentation as we had no previous indications. These lines have been changed as follows:
“Based on the outcomes of the experiments conducted in this research, it can be stated that, while certain models implemented with Keras exhibit low-performance classification, others represent a promising approach for the automated identification of these specific species, which are relevant to the preservation of the fauna of this archipelago.”
**********************************************************************
Comment 13
Line 497: What do you mean with "this type of species"?
---------------------------------------------------------------------------------------------------------
Answer to Comment 13
The authors have changed the third paragraph in Section 5 to clarify this issue:
“In addition, the research conducted in this study has demonstrated that certain implemented base models exhibit a more favourable trend in the classification of the species under study. Consequently, these models could be specifically considered in the development of a practical system for identifying these particular species. Notably, they have shown promising results even when subjected to samples from the Wild Dataset implying successful performance under adverse visibility conditions for the species. Although the images in this study were sourced from various internet platforms, the insights gained could be applied in future experiments using images captured with camera traps, given their similar visibility characteristics. Thus, the findings of this research hold promise for the development of monitoring systems based on camera traps for real-world applications.”
**********************************************************************
Comment 14
Lines 498-500: I think this is a speculative statement that is not backed by your actual results - your photos were not from camera traps.
---------------------------------------------------------------------------------------------------------
Answer to Comment 14
As it was mentioned, the authors have changed the third paragraph in Section 5. To emphasize this aspect, although it seems a feasible issue, it is a hypothesis. The next step would be to test with camera traps, but that part is more development than research.
**********************************************************************
Comment 15
Comments on the Quality of English Language
At times the wording is awkward and the style is just strange (e.g., "The study culminates with a comparison...", "...all of these are reptiles which are interesting to ecological concerns.", etc.). I think that the manuscript should be checked by a proficient English speaker.
---------------------------------------------------------------------------------------------------------
Answer to Comment 15
The authors have changed the wording of the paper.
"The study culminates with a comparison..." have been changed by:
“This study concludes with a comparison…”
"...all of these are reptiles which are interesting to ecological concerns." have been changed by:
“The selected reptile species included in the database are of ecological significance,”
**********************************************************************

Reviewer 3 Report

Comments and Suggestions for Authors

This study aims to identify endemic and invasive alien reptile species in the Canary Islands based on the deep learning models. Thirty-one convolutional neural network (CNN) models were applied to classify four reptile species including two endemic and two invasive alien species for developing a detection system based on automatic species recognition. While this study might introduce some novelty by applying deep learning techniques to detect and classify endemic and invasive alien species in the Canary Islands for the first time, the current manuscript should be improved according to the following comments.

In the study, 31 CNN models were used to classify four reptile species: two endemics (Gallotia stehlini and Chalcides sexlineatus) and two invasive alien species (Chamaeleo calyptratus and Python regius). However, the current manuscript lacks a sufficient explanation for the necessity of employing deep learning models to classify these four species in spite of these can be easily identified by the human eye. Therefore, the authors should clarify the rationale behind using deep learning for the classification of these species.

In addition, this study utilized only 40 samples per species obtained from public databases. The dataset seems insufficient in size for effectively training the deep learning models. Additionally, the models were trained with a maximum of 100 epochs setting, which is considered too limited for deep learning model training. Typically, deep learning models are trained with higher epochs, such as 500 or 1,000, along with implementing an early stop function to prevent overfitting. Despite the small size of the dataset and the low number of epochs used for training, the observed high accuracy of the model shown in the study might be attributed to the ease of classification for the four examined species, which can be accomplished even by human visual identification.양식의

Moreover, despite the assessment of 31 CNN models for classifying four reptile species, a comprehensive explanation of the study's findings lacks in the Results. Additionally, it seems to encompass only a concluding statement rather than a detailed discussion in the Discussion. The authors should add clearer and more comprehensive content in both the Results and Discussion for better elucidation of outcomes and implications of the study.양식의

Comments on the Quality of English Language

Minor editing of English language required

Author Response

**********************************************************************
Comment 1
In the study, 31 CNN models were used to classify four reptile species: two endemics (Gallotia stehlini and Chalcides sexlineatus) and two invasive alien species (Chamaeleo calyptratus and Python regius). However, the current manuscript lacks a sufficient explanation for the necessity of employing deep learning models to classify these four species in spite of these can be easily identified by the human eye. Therefore, the authors should clarify the rationale behind using deep learning for the classification of these species.
---------------------------------------------------------------------------------------------------------
Answer to Comment 1
Thank you for the useful comments.
The authors have added two paragraphs, the sixth and seventh in Section 1, to state the significance of the research problem and how an automated identification system can aid in the conservation efforts initiated by regional authorities.
The previous paragraphs (fourth and fifth paragraphs) are copied to contextualise this issue:
“The regional authorities have taken proactive measures to control the proliferation of invasive species, exemplified by the establishment of the Canary Islands Early Warning Network for the Detection and Intervention of Invasive Alien Species, known as RedEXOS (La Red de Alerta Temprana de Canarias para la Detección e Intervención de Especies Exóticas Invasoras) [5]. The management strategies employed rely on the information system designed for monitoring invasive alien species in the Canary Islands. This system functions as an administrative communication mechanism, hinging on the voluntary participation of individuals who report the presence of specimens when sightings occur. Nevertheless, the issue persists, as the population sizes of certain invasive species remain uncertain and it poses an ongoing threat to native species, as the invasive species continue to jeopardize the ecological balance.
Given the impossibility of precisely determining the times at which these species appear and recognizing that the warnings issued through the Canary Islands Early Warning Network rely on sporadic sightings by volunteers, from the Signals and Communications Department of the Las Palmas de Gran Canaria University, we consider the possibility of developing a detection system based on automatic species recognition using deep learning (DL) techniques, aiming to enhance the efficiency of monitoring and controlling the relevant species in the Canary Islands.”
The added paragraphs are as follows:
“To address the challenges posed by the absence of sightings or reported presence of relevant species, we propose the implementation of an automatic monitoring system. This system would utilize volumetric motion sensors and camera traps to detect and
record the presence of species more proactively and continuously. Volumetric motion sensors can be employed to activate camera traps upon detecting the presence of species in pre-defined spaces. The integration of these motion sensors with camera traps allows for the automated recording of images when triggered by the detected motion. This particular camera type provides the capability for continuous real-time monitoring. By incorporating automatic identification algorithms, the system can promptly activate an alert signal, notifying system administrators of the presence of any of the relevant species. The identification of species through strategically placed cameras in the Canary Islands will empower regional authorities to implement targeted measures. This includes actions like capturing the identified specimens and providing improved ecosystem monitoring.
To initiate the research, this paper suggests conducting an initial identification study focused on certain species of faunistic interest using various classification models. Two of these species are documented as having particular significance in the State Catalogue, while the other two are included in the Spanish Catalogue of invasive alien species.”
Later, more detailed information on these four species is provided in section 2.1.
**********************************************************************
Comment 2
In addition, this study utilized only 40 samples per species obtained from public databases. The dataset seems insufficient in size for effectively training the deep learning models.
---------------------------------------------------------------------------------------------------------
Answer to Comment 2
The authors of this study are aware that 40 samples is too low a number. The reason for using a dataset with so few samples is because the samples have been selected one at a time from internet searches.
In general, the images available on specialized websites usually present the photographed specimens in very favorable visibility conditions. However, finding images on the internet of species photographed in unfavorable conditions for identification (Wild Dataset) is a complicated task due to the scarcity of such images, especially for endemic species in the Canary Islands.
Nevertheless, despite the low number of samples available in this study, authors have considered it sufficient for this initial approach. Future work will increase both the number of samples and the number of classes, considering the models that have provided the best results, and adding modifications to the architecture that can improve performance.
**********************************************************************
Comment 3
Additionally, the models were trained with a maximum of 100 epochs setting, which is considered too limited for deep learning model training. Typically, deep learning models are trained with higher epochs, such as 500 or 1,000, along with implementing an early stop function to prevent overfitting.
---------------------------------------------------------------------------------------------------------
Answer to Comment 3
When the first simulations were carried out, the base model used did not reach 100 epochs, so it was therefore decided to apply this maximum epoch criterion for all models. According to what you state, the EfficientNet family models as well as the ResNet101 model reach the limit of 100 epochs meaning that if this limit had been increased, these models would have greater learning opportunities, and most likely, leading to better performance. However, the same parameter settings have been set for all base models in order to be able to make a fair comparison between the performance results obtained. Increasing the maximum number of epochs is one of the points considered in future lines of research (last paragraph of the discussion in Section 5).
On the other hand, the authors have added the following paragraph in section 3.1 (third from last paragraph) to clarify the stopping criteria in training:
“In these experiments, a maximum of 100 epochs and a patience set to 20 epochs have been defined for training each model. The objective of training is to minimize the loss. This metric is monitored at the end of each epoch and the training process concludes either when the loss no longer decreases after 20 epochs or when 100 epochs of training have been completed. The model weights are then restored to the weights from the best epoch in the training process.”
Regarding overfitting, the authors have used the method cross validation 5-fold, to avoid the overfitting.
**********************************************************************
Comment 4
Despite the small size of the dataset and the low number of epochs used for training, the observed high accuracy of the model shown in the study might be attributed to the ease of classification for the four examined species, which can be accomplished even by human visual identification.
---------------------------------------------------------------------------------------------------------
Answer to Comment 4
If the species under study were very similar to each other, the accuracy values would most likely not be so high. However, the comparison made in this initial approach is
useful to know which base models offer better performance and which ones offer poor performance, even though these species are very different from each other.
That is to say, InceptionV3 is the base model from which the lowest mean accuracy value is obtained in this study (29.37%). This information suggests that InceptionV3 can be discarded when designing a system that can be used in a real application, while the base models that offer better performance should be considered when designing more capable systems.
**********************************************************************
Comment 5
Moreover, despite the assessment of 31 CNN models for classifying four reptile species, a comprehensive explanation of the study's findings lacks in the Results. Additionally, it seems to encompass only a concluding statement rather than a detailed discussion in the Discussion. The authors should add clearer and more comprehensive content in both the Results and Discussion for better elucidation of outcomes and implications of the study.
---------------------------------------------------------------------------------------------------------
Answer to Comment 5
When this paper was structured, it was decided to divide the Results and Discussion section into two sections: the Results section mainly to present the results obtained in the different experiments, and the Discussion section to give an assessment of the results.
The authors have included a comprehensive explanation of the study's findings as a more in-depth discussion of the results in Discussion such as it can be seen in Section 5.
**********************************************************************
Comment 6
Comments on the Quality of English Language
Minor editing of English language required
---------------------------------------------------------------------------------------------------------
Answer to Comment 6
English language has been reviewed and therefore, the paper has been improved.
